**Data Availability Statement:** All relevant data are within the paper and its Supporting Information files.

**Funding:** This study was conducted with the financial support of the Africa Rising Project- a

# Volume estimation models for avocado fruit

**Mulugeta Mokria** [1]\*, **Aster Gebrekirstos**[2], **Hadia Said**[1], **Kiros Hadgu**[1], **Niguse Hagazi**[1], **Workneh Dubale**[3], **Achim Bräuning**[4]

**1** World Agroforestry (ICRAF), C/O ILRI Campus, Gurd Shola, Addis Ababa, Ethiopia, **2** World Agroforestry (ICRAF), United Nations Avenue, Nairobi, Kenya, **3** International Livestock Research Institute (ILRI), Addis Ababa, Ethiopia, **4** Institute of Geography, Friedrich-Alexander-University Erlangen-Nuremberg, Erlangen, Germany

\* m.mokria@cgiar.org, mgmokria@gmail.com

## Abstract

Avocado (*Persea americana* Mill.) is an important horticultural crop and proved to be a very profitable commercial crop for both local consumption and export. The physical characteristics of fruits are an important factor to determine the quality of fruit produced. On the other hand, estimation of fruit volume is time-consuming and impractical under field conditions. Thus, this study was conducted to devise cultivar-specific and generalized allometric models to analytically and non-destructively determine avocado fruit volume of five wildly distributed avocado cultivars. A significant relationship ($P \leq 0.01$) was found between fruit diameter, length, and volume of each cultivar. Our best models (VM2 –for cultivar specific, and VM7-generalized model) has passed all the rigorous cross-validation and performance statistics tests and explained 94%, 92%, 87%, 93%, 94% and 93% of the variations in fruit volume of Ettinger, Fuerte, Hass, Nabal, Reed, and Multiple cultivars, respectively. Our finding revealed that in situations where measurements of volume would be inconvenient, or time-consuming, a reliable volume and yield estimation can be obtained using site- and cultivar-specific allometric equations. Allometric models could also play a significant role in improving data availability on avocado fruit physical appearance which is critical to assess the quality and taste of fresh products influencing the purchase decision of customers. Moreover, such information can also be used as a ripeness index to predict optimum harvest time important for planned marketing. More importantly, the models might assist horticulturists, agronomists, and physiologists to conduct further study on avocado production and productivity through agroforestry landuse system across Ethiopia.

## Introduction

Avocado is a highly variable species and classified into three ecological races (i.e., the West Indian (WI), Guatemalan (G) and Mexican (M) "races") [1, 2]. It is an evergreen tree species and the most economically important species of the Lauraceae family. It is grown commercially in America, Africa, Europe, Asia and Oceania. In 2019, the estimated world's total avocado production was about 7.2 million tonnes from 726, 660 hectares [3]. The major avocado-growing countries are Mexico, USA, Colombia, Indonesia, Chile, the Dominican Republic, Kenya and South Africa [3]. In Africa, Kenya and South Africa are leading in the production

program financed by the United States Agency for International Development (USAID) as part of the United States Government's Feed the Future Initiative.

**Competing interests:** The authors have declared that no competing interests exist.

and export of avocado to the global market [4]. In Ethiopia, it was first introduced around 1938 in the eastern and southern parts of the country and it is now being widely distributed throughout the country, and mainly used for household consumption and local market [5–7]. Currently, Ethiopia is one of the top five avocado-producing countries in sub-Saharan Africa (SSA) and 20th in the world [3].

Avocado is recognized as a source of energy and vitamins. It also provides specific non-nutritive physiological benefits that may enhance health [8], thus, it can be considered as a "functional food" [9]. It is one of the top important commercial crops to be traded at a global scale [10, 11], and becoming one of the most promising fruit crops for both food and nutrition security and earning a considerable amount of financial return from export and domestic market [12–14]. Due to government initiatives in promoting investment in horticulture sector as well as combating climate change through land diversification and agroforestry practices, the plantation and the production of avocado is considerably increasing over the last few years across different parts of Ethiopia. Despite the expansion of avocado tree plantations, the physical characteristic of avocado fruits and productivity of small-scale avocado farming are not well studied in Ethiopia. On the other hand, information on fruit size is critical factor to determine the quality of the avocados and has been used to describe the fruit's growth curve, predict yield, and conduct physiological studies [15]. More importantly, on-tree and non-destructive volume estimates can be used as a ripeness index to predict maturity and optimum harvest time, and can be used while deciding packing material (tray insert) purchasing and marketing arrangements [16]. For physiological studies, measurement of the size of individual fruit over time allows monitoring fruit expansion rate and its response to physiological disorders and agronomic conditions [16, 17]. Moreover, in the context of postharvest operations, fruit size determination is important for several reasons, such as to determine packing material, fruit classified into batches of uniform size, assign market and price differentials of large and small produce, to match consumer preferences [18]. Thus, the availability of reliable fruit size information is critical in horticultural crop processing and marketing. Fruit volume is also a good measure of size, but direct measurement of fruit volume using water displacement approach is time-consuming as well as impractical under field conditions. On the other hand, length and width measurements of avocado fruit are quick and easy in the field or indoors and can be used to numerically represent fruit volume and weight. Therefore, this study was conducted to: (a) determine fruit volume of five avocado cultivars, (b) determine which biometric parameter of the avocado fruit best correlates with volume; (c) derive various cultivar-specific and mixed-cultivar allometric equations to predict fruit volume and (d) to evaluate the predictive performances of the equations and to identify the best allometric equation for the study region.

## Materials and methods

### Study area and climate characteristics

The study was conducted in the Upper Gana (7˚ 34' 24" N, 37˚ 46' 4"E) and Jewe (7˚ 30' 35" N, 37˚ 47' 1") Kebeles of Limu district, situated in Hadiya zone, in the Ethiopian Southern Nations, Nationalities, and Peoples' Region (SNNPR) [19] (Fig 1). The study area is located at 223 km South of Addis Ababa. The altitudinal ranges of Jewe and upper-Gana Kebles were between 2000 and 2400m (Fig 1). Based on the data from the Central Statistical Agency of Ethiopia (CSA), the district has an estimated total population of 153,783 and 93% of the people are living in the rural areas and practicing subsistence farming depending on rainfed production system [20]. The average farm size per family head was estimated to be 0.5 ha [19].

In the study area, the annual rainfall ranges between 1300 and 1400 mm with a bi-modal rainfall seasonality, occurring from February to April and from June to September [19]. The

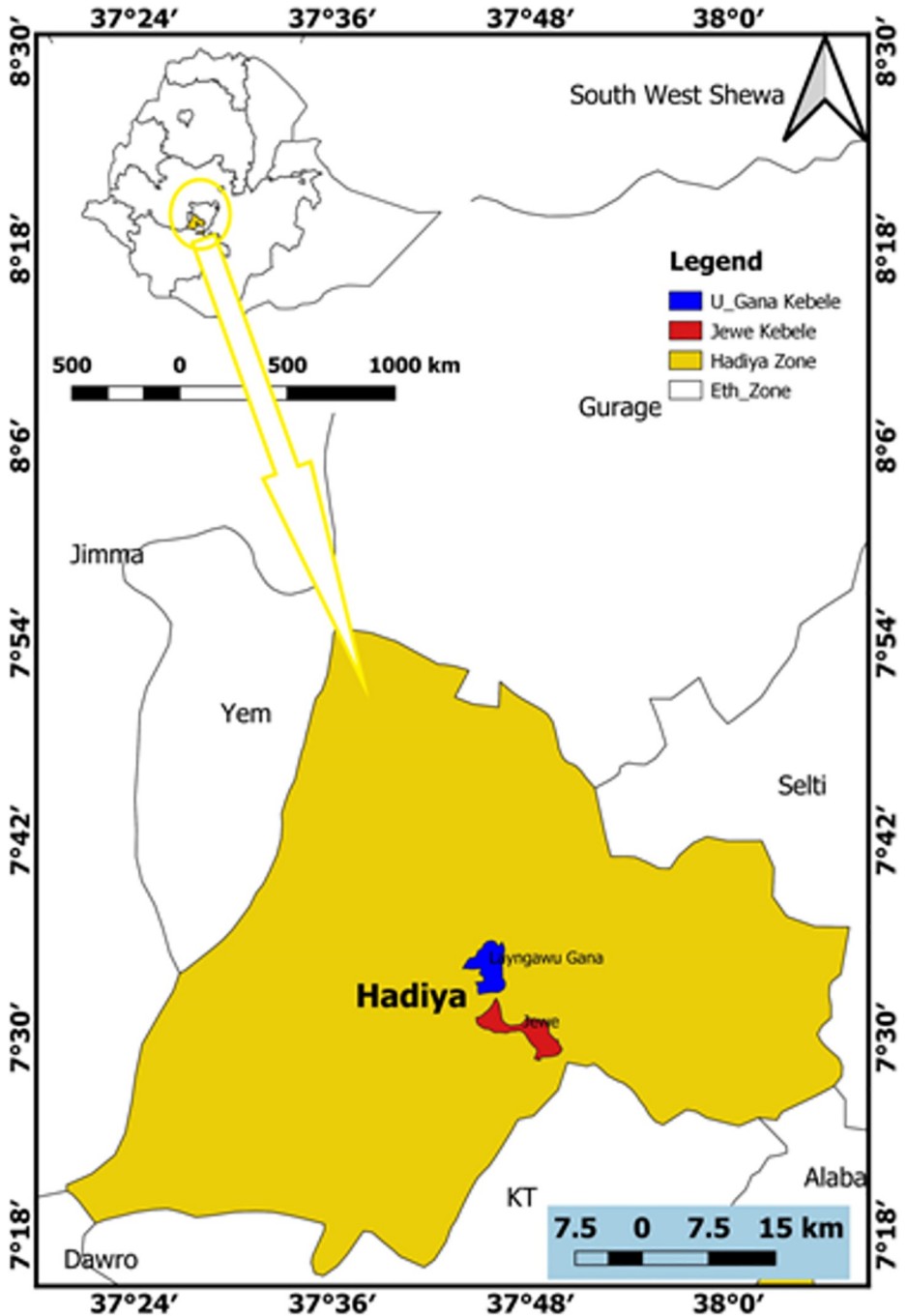

**Fig 1. The location of the study area in Hadiya zone, Ethiopia.** The shape files were accessed from open data sources of the open AFRICA https://open.africa/dataset/africa-shapefiles), http://geoportal.icpac.net/layers/geonode%3Aafr_g2014_2013_0 (Open access).

average annual minimum and maximum temperatures were 18˚C and 23˚C [19]. The typical land use system of the district is characterized by Agroforestry (i.e., mixed crop-tree-livestock production) [21]. The district has a favourable climate and agroecological condition for multi-strata agroforestry and home garden intensive farming system.

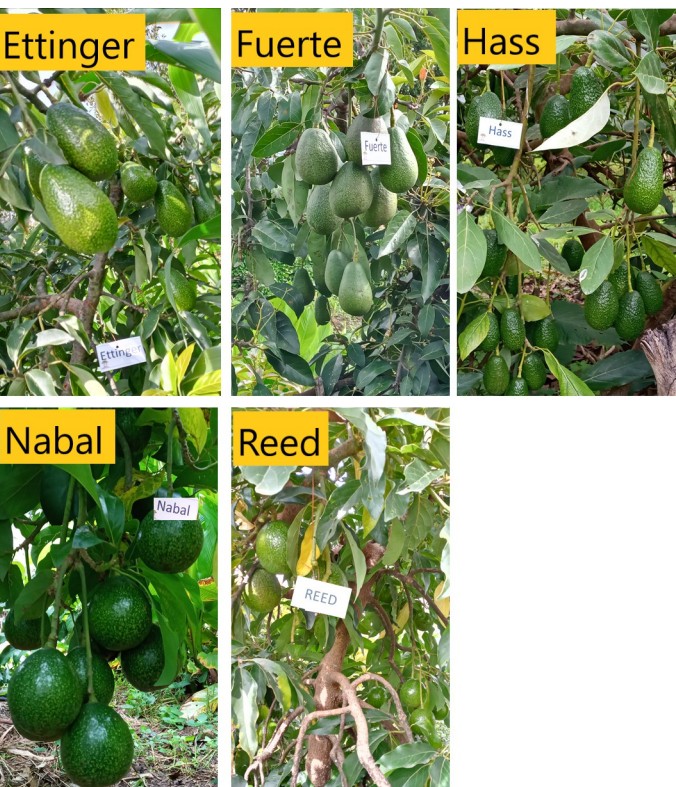

**Fig 2. Photographic documentation of the studied avocado cultivars during fruit sample collection.**

## Tree selection for sample fruit collection

In this study, five different avocado cultivars including Ettinger (race—Guatemalan (G) X Mexican (M) hybrid), Fuerte (race–G X M), Hass (race—G X M), Nabal (race—G), and Reed (race—G) were considered (Fig 2, S1 File) [22, 23]. From each cultivar, 30 avocado trees, representing 30 small-scale farmlands, were selected for fruit sample collection. Fruit samples were collected considering each radii of the crown [24–27]. A total of 360 fruit samples were randomly harvested from each cultivar (i.e., three fruits from each radius or 12 sample fruits per tree = 12 fruit X 30 trees = 360 fruits). Then, fruit length (FL in mm), diameter (FD in mm) and weight (FW in g) were measured using digital calliper and weighing scale, respectively. For fruit volume measurement, the collected sample fruits were first classified into three sizes (small, medium, and large). Then, 15 fruits were randomly selected from each size class (total = 45 fruits from each cultivar). For each fruit, actual fruit volume (AFV) was measured using water displacement method [15, 28], in the Biotechnology Laboratory of Wachemo University (WCU), Hossana, Ethiopia.

## Model development, performance evaluation and cross-validation test

Cultivar-specific and mixed-cultivar generalized avocado fruit volume (FV) estimation models were developed using linear and non-linear regression equations based on either fruit diameter, fruit length alone or both fruit length and diameter at the same time as independent variables. Moreover, using two predictors (i.e., length and diameter) may introduce potential problems of co-linearity, resulting in poor precision in the estimates of the corresponding regression coefficients [29]. Thus, we conducted a multicollinearity analysis using Variance

Inflation Factors (VIF = 1/(1-r$^2$)) and the Tolerance Values (T = 1/VIF) [30]; where r is the correlation coefficient between length and diameter of fruit. VIF value exceeding 10 or if T value was smaller than 0.10 then co-linearity may have a considerable impact on the prediction of the parameters, and consequently, one of those should be excluded from the model [31, 32].

Model performance was checked using various goodness-of-fit statistics, such as the Coefficient of Determination (R$^2$), Standard Error of Estimate (SEE), Index of Agreement (D), Mean Absolute Bias (MAB), Percent Bias (PBIAS), Root Mean Square Error (RMSE), Prediction Residuals Sum of Squares (PRESS), Reduction of Error (RE), and Coefficient Efficiency (CE), [33–35]. The estimation models with higher R$^2$ may sometimes also have unstable parameters estimates. Thus, we further calculated Percent Relative Standard Error (PRSE) of the coefficients and Weighted Akaike information criterion (AICiw) to check the stability of model parameter estimates [29]. Outlier and influential diagnostic test statistics, including Cook's distance, Leverage point, Studentized Residuals and DFFITS were analysed to examine the accuracy of model fit [29, 36]. Finally, models were evaluated and ranked based on all goodness-of-fit statistics, outlier, and influential diagnostic statistics [35].

To validate the best fitting equation for volume estimation, model cross-validation was conducted following a split-sample approach in which 45 measured fruit sample were partitioned into two sets, 33 for "training" (i.e., to develop the equations) and the remaining 12 fruit samples for "testing" the equations. The partitioning was performed according to the following procedure. From each size class (small, medium, and large) of the samples, four samples were randomly selected to perform the "test" dataset and the remaining sample fruits were used to form the "training" dataset. The goodness-of-fit statistics and equation coefficients of the "training" equations were compared with those derived using the full dataset, and the estimated and measured volume of "test" sample fruit was compared [35, 37, 38]. Finally, the full dataset was used to build fruit volume estimation models.

## Statistical analysis

Pearson correlation tests were conducted between actual AFV-FD and FL to be able to identify which fruit biometric variables were most strongly correlated with FV. A correlation analysis was also conducted between independent variables (i.e., FL vs FD). The differences among avocado cultivars in AFV was assessed using one-way analysis of variance and the significance of differences were tested using the least significant difference test (LSD) with $P < 0.05$ [35].

## Results and discussion

### The relationship of physical characteristics of avocado fruits

The physical characteristics of sample avocado fruits and their statistical attributes are presented in Table 1. The AFV ranged from 125–480 cm$^3$, FW ranged from 129 – 595g, FL ranged from 64.5–129.9 mm, and FD ranged from 53.8–99.8 mm (Table 1). These findings are in line with

**Table 1. Summary of volume, and fruit size characteristics of five avocado cultivars.**

| | Fruit Length (mm) | | Fruit Diameter (mm) | | Fruit wight (g) | | Fruit Volume (cm₃) | |
| --- | --- | --- | --- | --- | --- | --- | --- | --- |
| SPP name | Mean [SE] | Range | Mean [SE] | Range | Mean [SE] | Range | Mean [SE] | Range |
| Ettinger | 108.9 [±1.5] | 91.9–129.9 | 69.1 [±0.9] | 57.1–84.2 | 253 [±8.8] | 162–375 | 274.1 [±10.1] | 145–410 |
| Fuerte | 106.2 [±1.4] | 87.7–124.6 | 68.4 [±0.8] | 57.4–79.6 | 248.8 [±8.2] | 146–348 | 252.0 [±7.4] | 150–350 |
| Hass | 90.3 [±1.4] | 72.8–113.9 | 64.7 [±0.8] | 53.8–76 | 192.6 [±7.0] | 116–310 | 181.6 [±5.8] | 110–300 |
| Nabal | 92.6 [±1.5] | 70.1–118.7 | 82.6 [±1.2] | 65.8–99.8 | 331.9 [±13.8] | 160–595 | 324.7 [±11.3] | 180–480 |
| Reed | 81.3 [±1.3] | 64.5–98.1 | 74.1 [±1.1] | 60.8–89.2 | 242.3 [±10.3] | 129–426 | 247.1 [±10.4] | 125–420 |

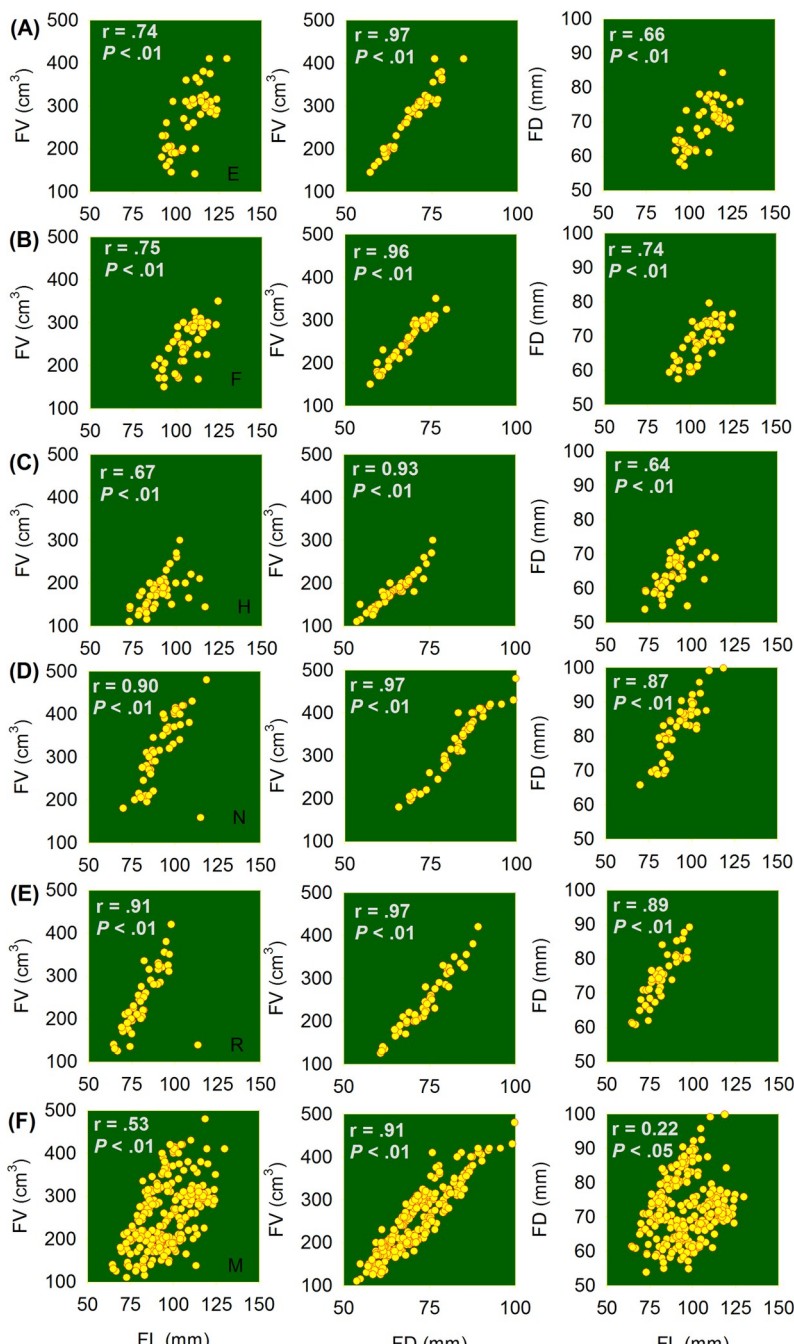

**Fig 3. Fruit volume as a function of fruit length, diameter, and regression of fruit diameter as a function of fruit length.** On Fig 3, the letter E, F, H, N, R and M refers Ettinger, Fuerte, Hass, Nabal, Reed and Mixed cultivar data.

other reports [2, 39, 40]. Information on fruit physical appearance is important for the customer who is used to assess the quality and taste of fresh product influencing the purchase decision [16].

The relationship between FV and other pomological traits (i.e., FL, FD,) were significant ($P < 0.01$) (Fig 3) and that is in agreement with other studies that have shown strong relationships between different pomological traits of avocado fruit [2]. Moreover, the correlation between FD and FV were consistently stronger compared to the relationship between FL and

FV (Fig 3), indicating the FD is a very good predictor of fruit volume and highlighting those changes in diameter, therefore, affects the fruit volume more than does a change in fruit length. More importantly, knowing the relationship between fruit physical characteristics is critical in the horticultural sector because these pomological traits are sometime used as fruit maturity index [2, 41, 42]. Several indices have been used to determine avocado fruit maturity, hence there is no single factor can be considered the most important; however, it can be stated that from a postharvest standpoint, quality begins at harvest with physiological maturity. This indicates that understanding the stage of physiological maturity is critical for the development of a successful avocado fresh fruit industry while assuring the quality to the consumer [43].

## Allometric equations and model cross-validation

As a preliminary step to model calibration, the degree of collinearity among fruit length and diameter was analysed. The VIF was ranged from 1.1 to 4.6 and T values ranged from 0.21 to 0.95, depending on cultivar type, respectively (S1 Table). Hence, for all selected genotypes, VIF was < 10 and T was > 0.10, showing that the co-linearity between predictors (fruit length and diameter) is negligible, thus both predictors (FL, FD) were considered during model formation [31, 32] (S1 Table).

Cultivar-specific and mixed cultivar generalized volume model was developed to approximate the shape of avocado fruits using either fruit length (FL) or width (FD) separately and using both predictors at a time. We found that the predictive performance of tested model form were varied within cultivar (Table 2, S1 Table). This might be attributed to differences in the equation forms and predictors included in the models [35]. Allometric model performance analysis and cross-validation test results showed that the Linear Regression Model (MV2) which includes only FD as predictor was ranked the best model given the set of nine (9) candidate model forms for all cultivar-specific models, while the Multiple Linear Regression Model (VM7) was the best for generalized mixed cultivar model (Table 2), and detailed model performance statistics for all tested model forms are presented in S1 Table).

Our best model explained 94%, 92%, 87%, 93%, 94% and 93% of variation in fruit volume in Ettinger, Fuerte, Hass, Nabal, Reed and mixed-avocado cultivar model, respectively (Fig 4, Table 2, in S1 Table for more details). The three best-performing models for each avocado cultivar and mixed-cultivar are shown in Table 2, where the influence of coefficients was significant ($P < 0.01$) (Table 2). Our best model (VM2 for cultivar specific) and VM7 –for mixed cultivar models have passed all the rigorous verification and cross-validation statistical test and produced the lowest average relative error (PBIAS%), implying that fruit diameter is reliable predictors of cultivar specific fruit volume, while using both FL and FD might increase the predictive performances of generalized allometric models (S1 Table). Moreover, the performance of our best models (VM2, and VM7) to make an accurate prediction is not an artifact of overfitting, because the parameter values were stable across the subset of the cross-validations "training data set" and full data set (S1 Table). Moreover, the volume of the twelve "test" fruit volume estimated with the cross-validation equations (i.e., training dataset) differed little from the values estimated with equations produced with the full dataset (Fig 5, S1 Table). The deviations (PBIAS%) in the volume estimates between the two sets of equations were less than 1% for all and mixed cultivar equations (Fig 5). Moreover, the PRSE value of < 20% and outliers and influential points of less than 10% of the total observation as well as higher positive value of CE, and RE provides evidence that the parameter estimates were reliable in the selected best models [29, 44–46] (S1 Table). Thus, VM2 (i.e., cultivar-specific model) and VM7 (i.e., generalized model) are reliable to determine the fruit volume based on their easily measurable fruit pomological traits (i.e., FL or/and FD). Furthermore, measuring fruit length and diameter are

**Table 2. Equations and goodness-of-fit performance statistics for estimating avocado fruit volume of five different cultivars and multiple cultivars grown in Limo district, Hadiya, zone.**

| Model forms | Model forms | Coefficient | | | Performance statistics | | | | | | | | | | | | PRSE | | | Rank |
|---|---|---|---|---|---|---|---|---|---|---|---|---|---|---|---|---|---|---|---|---|
| | | a | b | c | R2 | SEE | PRESS | RMSE | PBIAS | MAB | Di | RE | CE | AICi | Δi(AIC) | Wi(AIC) | a | b | c | |
| **Ettinger** | | | | | | | | | | | | | | | | | | | | |
| VM2 | a*FD + b | 10.4484*** | -447.701*** | | 0.94 | 17.53 | 13210.88 | 17.13 | 0.00 | 12.20 | 0.98 | 1.00 | 0.94 | 259.70 | 11.31 | 0.00 | 4.0 | -6.5 | | **1** |
| VM4 | a*FD^2 | 0.0575*** | | | 0.88 | 24.19 | 25752.29 | 23.92 | -0.94 | 18.94 | 0.96 | 0.99 | 0.88 | 287.73 | 39.34 | 0.00 | 1.2 | | | **2** |
| VM9 | a*(FL*FD)^2 | 4.53E-06*** | | | 0.74 | 35.17 | 54412.87 | 34.77 | 2.71 | 28.37 | 0.95 | 0.98 | 0.74 | 321.40 | 73.01 | 0.00 | 1.9 | | | **3** |
| **Fuerte** | | | | | | | | | | | | | | | | | | | | |
| VM2 | a*FD + b | 8.369*** | -320.756*** | | 0.92 | 14.81 | 9425.88 | 14.47 | 0.00 | 10.93 | 0.98 | 1.00 | 0.92 | 244.50 | 0.22 | 0.46 | 4.6 | -8.3 | | **1** |
| VM4 | a*FD^2 | 0.0536*** | | | 0.89 | 16.33 | 11733.34 | 16.15 | -0.31 | 12.71 | 0.97 | 1.00 | 0.89 | 252.36 | 8.07 | 0.01 | 0.9 | | | 2 |
| VM7 | a+ b*FL + c*FD | -333.124*** | 0.5107 | 7.757*** | 0.92 | 14.62 | 8973.05 | 14.12 | 0.00 | 10.62 | 0.98 | 1.00 | 0.92 | 244.29 | 0.00 | 0.52 | -8.3 | **68.5** | 7.3 | 3 |
| **Hass** | | | | | | | | | | | | | | | | | | | | |
| VM2 | a*FD + b | 6.5239*** | -240.661*** | | 0.87 | 14.33 | 8832.76 | 14.01 | 0.00 | 9.00 | 0.96 | 0.99 | 0.87 | 306.82 | 69.78 | 0.00 | 5.8 | -10.3 | | **1** |
| VM4 | a*FD^2 | 0.0432*** | | | 0.86 | 14.65 | 9438.43 | 14.48 | -0.41 | 10.30 | 0.96 | 0.99 | 0.86 | 237.04 | 0.00 | 0.71 | 1.2 | | | 2 |
| VM7 | a+ b*FL + c*FD | -252.658*** | 0.5479 | 5.945*** | 0.88 | 13.93 | 8149.19 | 13.46 | 0.00 | 9.03 | 0.97 | 0.99 | 0.88 | 239.96 | 2.91 | 0.17 | -9.9 | **53.3** | 8.1 | 3 |
| **Nabal** | | | | | | | | | | | | | | | | | | | | |
| VM2 | a*FD + b | 9.314*** | -444.277*** | | 0.93 | 20.20 | 17549.35 | 19.75 | 0.00 | 14.44 | 0.98 | 1.00 | 0.93 | 272.47 | 7.34 | 0.02 | 5.57 | -7.13 | | **1** |
| VM4 | a*FD^2 | 0.0475*** | | | 0.90 | 24.42 | 26241.01 | 24.15 | -0.63 | 20.08 | 0.97 | 0.99 | 0.90 | 288.58 | 23.45 | 0.00 | 1.05 | | | 2 |
| VM1 | a*FL + b | 6.8725*** | -311.718*** | | 0.80 | 34.55 | 51343.88 | 33.78 | 0.00 | 27.55 | 0.94 | 0.99 | 0.80 | 320.78 | 55.65 | 0.00 | 7.55 | -15.51 | | 3 |
| **Reed** | | | | | | | | | | | | | | | | | | | | |
| VM2 | a*FD + b | 9.3255*** | -443.701*** | | 0.94 | 17.99 | 13912.04 | 17.58 | 0.00 | 13.52 | 0.98 | 1.00 | 0.94 | 262.02 | 9.72 | 0.01 | 4.0 | -6.2 | | **1** |
| VM4 | a*FD^2 | 0.0453*** | | | 0.86 | 25.95 | 29637.53 | 25.66 | -1.56 | 21.66 | 0.95 | 0.99 | 0.86 | 294.06 | 41.76 | 0.00 | 1.5 | | | 2 |
| VM1 | a*FL + b | 7.0449*** | -325.924*** | | 0.83 | 29.02 | 36221.43 | 28.37 | 0.00 | 22.30 | 0.95 | 0.99 | 0.83 | 305.08 | 52.79 | 0.00 | 6.8 | -12.0 | | 3 |
| **Mixed** | | | | | | | | | | | | | | | | | | | | |
| VM7 | a+ b*FL + c*FD | -435.118*** | 1.8844*** | 7.110*** | 0.93 | 20.23 | 90851.70 | 18.84 | 0.00 | 15.44 | 0.98 | 0.99 | 0.93 | 1327.16 | 0.00 | 1.00 | -3.0 | 5.2 | 2.2 | **1** |
| VM2 | a*FD + b | 7.7671*** | -301.595*** | | 0.82 | 32.86 | 240785.37 | 29.00 | 0.00 | 27.08 | 0.95 | 0.99 | 0.82 | 1519.21 | 192.04 | 0.00 | 3.1 | -5.8 | | 2 |
| VM4 | a*FD^2 | 0.0491*** | | | 0.81 | 33.77 | 255523.93 | 30.18 | -0.41 | 27.80 | 0.94 | 0.98 | 0.81 | 1535.16 | 208.00 | 0.00 | 0.8 | | | 3 |

*** **is** significant at P<0.001. Bold PRSE values indicates unreliable parameter estimates.

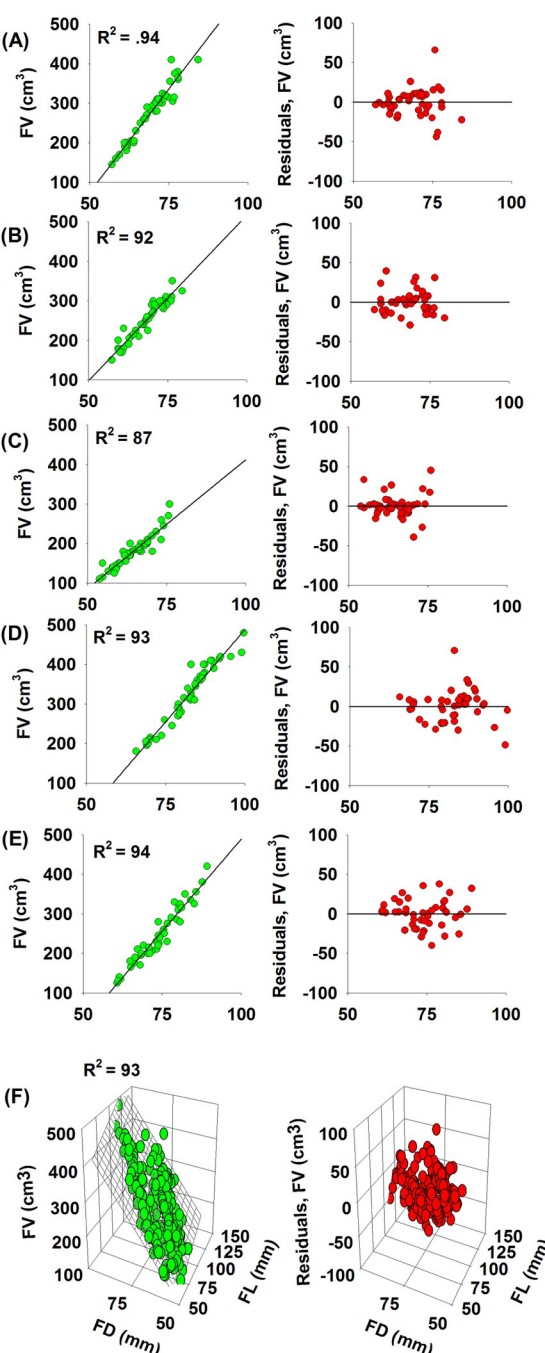

**Fig 4.** Relationships between FV and FD (*left panel*) and corresponding residual plots (*right*). Figs A-F refers Ettinger (A), Fuerte (B), Hass (C), Nabal (D), Reed (E) and Mixed species (F).

easy in the field, thus site-and cultivar-specific allometric model would enable researchers to make non-destructive or repeated measurements on the same fruits. Our allometric models could provide accurate estimate of avocado fruit volume and may reduce required time and financial resources while using common method of volume measurements like water displacement, gas displacement and expensive instruments, e.g., image processing software or machine vision techniques. In line with this, review literature showed that there are different non-

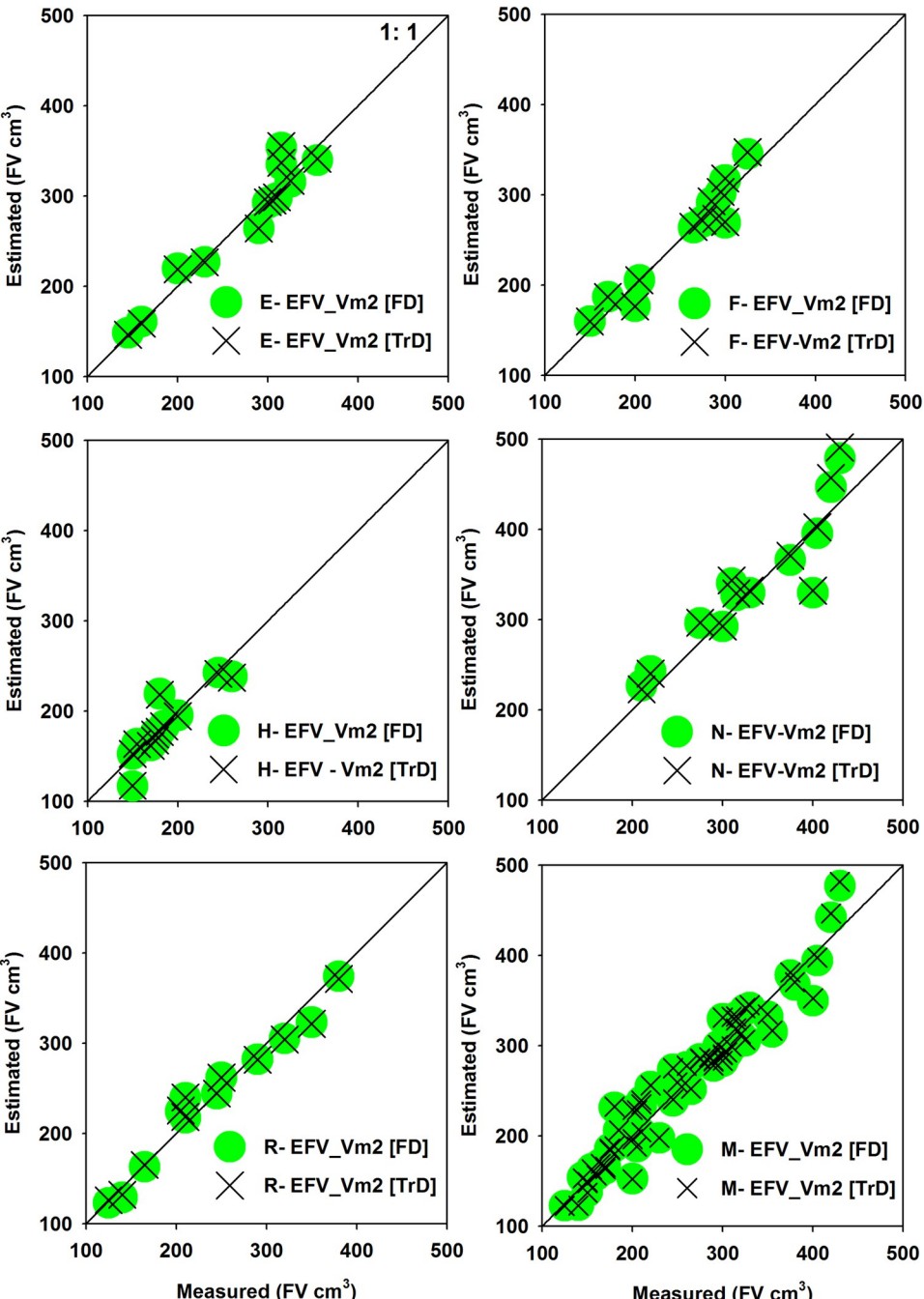

**Fig 5. Relationship between estimated and measured fruit volume of the 12 cross-validation test fruits.** Circles are the volume estimates calculated using the full data set equations and the crosses are the estimates calculated using the cross-validation training dataset equations. On Fig 5, the letter E, F, H, N, R and M refers Ettinger, Fuerte, Hass, Nabal, Reed and Mixed cultivar data.

destructive volume estimation models developed from easily measurable parameters for different type of fruits such as pepper [15, 17], Babassu (*Attalea speciosa*) fruit [47], Karanda (*Carissa carandas*) fruit [31], and Apple fruit [48]. Our findings confirmed that non-destructive allometric models based on easily measurable morphometric dimensions can be more accurate

and practical in field conditions than destructive methods used in traditional growth curves [18].

## Conclusions and recommendation

This study provided the first avocado cultivar-specific and mixed-cultivar generalized allometric equations to estimate avocado fruit volume non-destructively. Among tested model forms, VM2 (for cultivar-specific model), and VM7 (generalized model) have passed all rigorous verification and cross-validation statistical tests, confirming that the models have sufficient skill to estimate fruit volume from easily measurable parameters. Our best allometric models (VM2 and VM7) explained > 87% of the variation in measured fruit volumes of each cultivar. A high degree of correlation ($R^2 > 0.93$) between measured and estimated fruit volume provided quantitative evidence of the validity of the selected volume estimation models. The allometric equation developed in this research could be practical in the estimation of avocado fruit volume and applicable under field conditions. Besides, the generalized mixed-cultivar model can reliably be used to estimate avocado fruit volume when cultivar type is unknown. Our finding revealed that in the situations where fruit length and diameter measurements are possible and/ or where measurements of volume would be inconvenient, or time-consuming, site- and cultivar-specific allometric equations can be used to estimate fruit volume while it is on the tree. Therefore, the allometric equations generated in this study could play a considerable role in improving data availability on avocado fruit physical appearance which is critical to assess the quality and taste of fresh products, which in turn, influences the purchase decision of customers. It can also potentially assist horticulturists, agronomists, and physiologists to estimate fruit volume of avocado accurately and to carry out yield estimation before harvesting. Finally, we also recommend conducting a similar study using a large dataset collected from different agro-ecological regions, which would help to have a robust generalized and species-specific avocado volume estimation model that can be used across regions.

## Supporting information

**S1 File. Colored photograph of five Avocado cultivars.**
(PDF)

**S1 Table. Model performance statistics.**
(XLSX)

## Acknowledgments

We thank the local community and local administration offices in the study area for their support during fieldwork. We are also grateful to Mr. Eyuel Tesfaye, and Mr. Demeke Beyen for their facilitation of the fieldwork and organizing the team of experts for data collection. The contents of this document are solely the responsibility of the author/s and do not necessarily represent the official views of USAID or the U.S. Government or that of the institutional position of ICRAF (World Agroforestry).

## Author Contributions

**Conceptualization:** Mulugeta Mokria, Aster Gebrekirstos, Kiros Hadgu, Niguse Hagazi, Achim Bräuning.

**Data curation:** Mulugeta Mokria, Hadia Said, Niguse Hagazi, Workneh Dubale.

**Formal analysis:** Mulugeta Mokria.

**Funding acquisition:** Aster Gebrekirstos.

**Supervision:** Achim Bräuning.

**Writing – original draft:** Mulugeta Mokria.

**Writing – review & editing:** Mulugeta Mokria, Aster Gebrekirstos, Hadia Said, Kiros Hadgu, Niguse Hagazi, Achim Bräuning.

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
