## [Decision Letter · Decision Letter 0]

10 Dec 2021

PONE-D-21-36255Volume estimation models for tropical fruitPLOS ONE

Dear Dr. Mokria,

Thank you for submitting your manuscript to PLOS ONE. After careful consideration, we feel that it has merit but does not fully meet PLOS ONE’s publication criteria as it currently stands. Therefore, we invite you to submit a revised version of the manuscript that addresses the points raised during the review process.

We look forward to receiving your revised manuscript.

Kind regards,

Sajid Ali

Academic Editor

PLOS ONE

3. Please ensure that you refer to Figure 4 in your text as, if accepted, production will need this reference to link the reader to the figure.

Reviewers' comments:

**Reviewer #1:** Dear,

This study provided the first avocado cultivar-specific and mixed-cultivar generalized allometric equations to estimate avocado fruit volume non-destructively.

Major comments:

1. The title of the article does not provide a clear picture of the subject under consideration. The title implies that a general allometric model for all tropical crops is provided, but this model was only used for non-destructive measurement of avocado fruit volume. As a result, it is suggested that the article be given a more accurate title.

2. Avocado is a highly variable species and three botanical subspecies or ecological races of P. americana Mill. were recognized: Mexican (M) (var. drymifolia), Guatemalan (G) (var. guatemalensis), and West Indian (WI) (Antillean) (var. americana). Cultivars derived from Guatemalan and Mexican races and their hybrids are grown primarily in subtropical climates and have physiological adaptations to cooler temperatures, as opposed to cultivars derived from West Indian race or WI­hybrids, which are adapted to tropical climates.

Regarding this horticultural fact, there must be consistency throughout the manuscript's text as well as title. Now, none of the cultivars studied in this study are of tropical origin.

Using the words 'tropical' in the title and line 17 and 'subtropical' in line 40 made inconsistencies that must be addressed.

3. According to line 117: “model performance was checked using various goodness-of-fit statistics, such as the Coefficient of Determination (R2), Standard Error of Estimate (SEE), Index of Agreement (D), Mean Absolute Bias (MAB), Percent Bias (PBIAS), Root Mean Square Error (RMSE), Prediction Residuals Sum of Squares (PRESS), Reduction of error (RE), and Coefficient efficiency (CE)”.

As is obvious, the regression was run separately for the actual fruit volume and each of the independent variables, such as fruit length, fruit weight, and fruit diameter. In other words, a single factor regression has been investigated (Length with volume, diameter with volume and weight with volume). As you are aware, mathematical relationships such as regression between the sum of length and diameter with volume, regression between the multiplication of length and diameter by volume, or regression between the square of length + diameter by volume are preferable. It is strongly advised to provide better estimation by providing more mathematical relationships between fruit length and diameter and regressing them with fruit volume so that a better choice between them can be made.

4. According to section 2.3: “Cultivar-specific and mixed-cultivar generalized avocado fruit volume (FV) estimation models were developed using linear and non-linear regression equations based on either fruit diameter, fruit length alone or both fruit length and diameter at the same time as ndependent variables”.

Please include both linear and non-linear regression equations in the table, as well as the Goodness of Fit for the linear equation. If a non-linear equation is used, please explain it using terms such as logarithm, polynomial, and so on. Non-linear regression can provide more accurate predictions and is especially valuable for biological data. The value and strength of the proposed model are increased by including these items.

5. According to line 128: “To validate the best fitting equation for volume estimation, model cross-validation was conducted following a split-sample approach in which 45 measured fruit sample were partitioned into two sets, 33 for ‘‘training’’ (i. e., to develop the equations) and the remaining 12 fruit samples for ‘‘testing’’ the equations”.

Twelve fruits from each cultivar were used for testing (validation), which appears insufficient and represents a small statistical population. Moreover, validation requires sampling from other gardens or areas in the same climate. In other words, performing large-scale sampling will result in strong validation with a high correlation coefficient.

6. After obtaining the model and estimating the data, regression validation should be performed to determine whether a suitable correlation coefficient exists. Please include graphs of validation-related correlations.

Minor comments:

1. Line 17: Mill -> Mill (non-italicized)

2. Line 17: remove "genus Persea" at the end.

3. Line 19: The physical characteristic -> The physical characteristics

4. Line 23: "five wildly distributed avocado verities" -> Q: five varieties or cultivars? As mentioned in the materials and methods section, they are 5 avocado cultivars.

5. Line 23: "five wildly distributed avocado verities" -> … varieties

6. Line 24: found between Fruit diameter -> fruit

7. Line 58: small scall avocado farming -> scale

8. Line 59: information of fruit size are critical factors -> is critical

9. Line 73: five avocado verities -> … varieties (Q: five varieties or cultivars?)

10. Line 77: Materials and methods -> Materials and Methods

11. Line 78: 2.1 -> 2.1.

12. Line 101: classified in to three size -> into

13. Line 130: i. e. -> i.e.

14. Line 139: Fruit volume -> fruit volume

15. Line 145: Result and discussion -> Results and Discussion

16. Line 234: (VM2 and VM2) -> ?

17. Line 312: reference # 18: ?

Other comments:

1. The main body and reference section must be adjusted in accordance with the author's guidelines.

2. The manuscript must be significantly improved and rewritten as a result of comments and suggestions. All of the issues raised should be addressed. The manuscript must then be reassessed for quality and suitability for publication in PLOS ONE.

**Reviewer #2:** 

Manuscript presents allometric models to non-destructively predict avocado fruit volume under both cultivar-specific and mixed-cultivar production systems. Introduction, Materials and Methods, and Results and Discussion sections are adequately written and justify the claims made in the manuscript. Data collection method, and statistical tests sufficiently verify the models. I suggest to maturity level of fruit at the time of harvest and add a colored photograph showing morphological differences among avocado fruits from studied cultivars. Since, all models have some limitations, it would be appropriate to add limitations for these models, too. Some English language improvements are suggested in the reviewed version of the manuscript.

---

## [Author Response · Author response to Decision Letter 0]

18 Dec 2021

Response to reviewers comment

ID: PONE-D-21-36255

Title: Volume estimation models for tropical fruit

Dear Editor 

First, we would like to express our sincere appreciation to you and the anonymous reviewers for their very insightful and constructive comments and suggestions to our manuscript. Kindly find the response to each of the comments below. We believe we were able to address all suggestions and comments adequately, otherwise please contact us.

Response to the editor: We are very great full for your comment and suggestion. We amended the structure/style of the revised manuscript as the journal guideline. Regarding, fig 1, we reproduced the map by taking the shapefiles from openAFRICA ( open sources Africa Shapefiles, please see: -https://open.africa/dataset/africa-shapefiles); http://geoportal.icpac.net/layers/geonode%3Aafr_g2014_2013_0.

On behalf of all co-authors,

Sincerely,

Mulugeta Mokria

Response to reviewer #1

This study provided the first avocado cultivar-specific and mixed-cultivar generalized allometric equations to estimate avocado fruit volume non-destructively.

Major comments:

Comment #1. The title of the article does not provide a clear picture of the subject under consideration. The title implies that a general allometric model for all tropical crops is provided, but this model was only used for non-destructive measurement of avocado fruit volume. As a result, it is suggested that the article be given a more accurate title.

Response #1: We thank the reviewer for making a very important point and we revised the title to “Volume estimation models for avocado fruit” please see line 1 in the revised manuscript. 

Comment #2. Avocado is a highly variable species and three botanical subspecies or ecological races of P. americana Mill. were recognized: Mexican (M) (var. drymifolia), Guatemalan (G) (var. guatemalensis), and West Indian (WI) (Antillean) (var. americana). Cultivars derived from Guatemalan and Mexican races and their hybrids are grown primarily in subtropical climates and have physiological adaptations to cooler temperatures, as opposed to cultivars derived from West Indian race or WI-hybrids, which are adapted to tropical climates.

Regarding this horticultural fact, there must be consistency throughout the manuscript's text as well as title. Now, none of the cultivars studied in this study are of tropical origin.

Using the words 'tropical' in the title and line 17 and 'subtropical' in line 40 made inconsistencies that must be addressed.

Response #2: The reviewer makes a very important point, and we are grateful for that. Thus, we amended the titles and avoided the inconsistency in using the term “Tropical” through the manuscripts. Please see lines 1, 17, 38 in the revised manuscript. 

Comment #3. According to line 117: “model performance was checked using various goodness-of-fit statistics, such as the Coefficient of Determination (R2), Standard Error of Estimate (SEE), Index of Agreement (D), Mean Absolute Bias (MAB), Percent Bias (PBIAS), Root Mean Square Error (RMSE), Prediction Residuals Sum of Squares (PRESS), Reduction of error (RE), and Coefficient efficiency (CE)”.

As is obvious, the regression was run separately for the actual fruit volume and each of the independent variables, such as fruit length, fruit weight, and fruit diameter. In other words, a single factor regression has been investigated (Length with volume, diameter with volume and weight with volume). As you are aware, mathematical relationships such as regression between the sum of length and diameter with volume, regression between the multiplication of length and diameter by volume, or regression between the square of length + diameter by volume are preferable. It is strongly advised to provide better estimation by providing more mathematical relationships between fruit length and diameter and regressing them with fruit volume so that a better choice between them can be made.

Response #3: The reviewer makes a very important point, and we are grateful for that. As hinted by the reviewer, we have tested about 9 model forms (see the supplementary information - S2_Model Performance statistics.xls) and we selected the three most potential models and their estimation efficiency also depends on the varieties. Please see the revised manuscript. 

Comment #4. According to section 2.3: “Cultivar-specific and mixed-cultivar generalized avocado fruit volume (FV) estimation models were developed using linear and non-linear regression equations based on either fruit diameter, fruit length alone or both fruit length and diameter at the same time as independent variables”.

Please include both linear and non-linear regression equations in the table, as well as the Goodness of Fit for the linear equation. If a non-linear equation is used, please explain it using terms such as logarithm, polynomial, and so on. Non-linear regression can provide more accurate predictions and is especially valuable for biological data. The value and strength of the proposed model are increased by including these items.

Response #4: We thank for the suggestions. We provided both linear and non-linear regression equations and with their goodness of fit statistics (Please see the Supplementary information- S2_Model Performance statistics.xls). However, for simplicity, we provided only the three most performed equations in the main text and supplied the statistical performances of all tested models in the supplementary information document with more explanation (S2_Model Performance statistics.xls). 

Comment #5. According to line 128: “To validate the best fitting equation for volume estimation, model cross-validation was conducted following a split-sample approach in which 45 measured fruit sample were partitioned into two sets, 33 for ‘‘training’’ (i. e., to develop the equations) and the remaining 12 fruit samples for ‘‘testing’’ the equations”.

Twelve fruits from each cultivar were used for testing (validation), which appears insufficient and represents a small statistical population. Moreover, validation requires sampling from other gardens or areas in the same climate. In other words, performing large-scale sampling will result in strong validation with a high correlation coefficient.

Response #5: The reviewer makes a very important point. We share the concern of the reviewers’ and agree on model validation using large and independent data will result greater confidence on model selection and use. However, we could not find such independent data in Ethiopia, and we also collected only 45 sample fruit from each variety. There for, we indicated as the limitation of the manuscript and suggested the need for further study using large database (if available). Please see lines 250 -252, in the revised manuscript. 

Comment #6. After obtaining the model and estimating the data, regression validation should be performed to determine whether a suitable correlation coefficient exists. Please include graphs of validation-related correlations.

Response #6: We thanks the review for pointing out this and we have provided a cross-validation graph. please see line 225, Fig 5, in the revised manuscript. 

Minor comments:

Comment #1. Line 17: Mill -> Mill (non-italicized)

Response #1: We revised the text accordingly, please see line 17 in the revised manuscript. 

Comment #2. Line 17: remove "genus Persea" at the end.

Response #2: We removed the word “genus Persea, please see line 17”in the revised manuscript. 

Comment #3. Line 19: The physical characteristic -> The physical characteristics

Response #3: Thanks, we revised the text accordingly, please see line 18 in the revised manuscript. 

Comment #4. Line 23: "five wildly distributed avocado verities" -> Q: five varieties or cultivars? As mentioned in the materials and methods section, they are 5 avocado cultivars.

Response #4: We thanks the review for pointing out this and changed the text to “cultivars” through the manuscript, please see line 23 in the revised manuscript. 

Comment #5. Line 23: "five wildly distributed avocado verities" -> … varieties

Response #5: Thanks, corrected accordingly, please see line 23 in the revised manuscript 

Comment #6. Line 24: found between Fruit diameter -> fruit

Response #6: We revised the text accordingly, please see line 23 in the revised manuscript. 

Comment #7. Line 58: small scall avocado farming -> scale

Response #7: Thanks, corrected accordingly, please see line 55 in the revised manuscript. 

Comment #8. Line 59: information of fruit size are critical factors -> is critical

Response #8: Corrected, please line 56 in the revised manuscript 

Comment #9. Line 73: five avocado verities -> … varieties (Q: five varieties or cultivars?)

Response #9: Thanks, changed to cultivars through the manuscript, please see line 70 in the revised manuscript. 

Comment #10. Line 77: Materials and methods -> Materials and Methods

Response #10: We revised the text accordingly, please see line 74 in the revised manuscript. 

Comment #11. Line 78: 2.1 -> 2.1.

Response #11: Thanks, revised accordingly, please see line 75 in the revised manuscript. 

Comment #12. Line 101: classified in to three size -> into

Response #12: We revised the text accordingly, please see line 101 in the revised manuscript

Comment #13. Line 130: i. e. -> i.e.

Response #13: Thanks, corrected accordingly, please see line 133 in the revised manuscript

Comment #14. Line 139: Fruit volume -> fruit volume

Response #14: We revised the text accordingly, please see line 140 in the revised manuscript

Comment #15. Line 145: Result and discussion -> Results and Discussion

Response #15: We revised the text accordingly, please see line 147 in the revised manuscript 

Comment #16. Line 234: (VM2 and VM2) -> ?

Response #16: Corrected and VM2 is changed to VM7, please see line 235 in the revised manuscript. 

Comment #17. Line 312: reference # 18: ?

Response #17: We thank the reviewer for pointing out this, and corrected it accordingly, please lines 314-15, in the revised manuscript. 

Other comments:

Comment #1. The main body and reference section must be adjusted in accordance with the author's guidelines.

Response #1: We thank the reviewer for pointing out this. We corrected the main body and the reference style as per the journal user guide. 

Comment #2. The manuscript must be significantly improved and rewritten as a result of comments and suggestions. All of the issues raised should be addressed. The manuscript must then be reassessed for quality and suitability for publication in PLOS ONE.

Response #2: We thank the review for constructive comments and suggestions to our manuscript. We believe we were able to include all suggestions and comments in an adequate way, otherwise please contact us.

Response to reviewer #2

Reviewer #2: 

Manuscript presents allometric models to non-destructively predict avocado fruit volume under both cultivar-specific and mixed-cultivar production systems. Introduction, Materials and Methods, and Results and Discussion sections are adequately written and justify the claims made in the manuscript. Data collection method, and statistical tests sufficiently verify the models. I suggest to maturity level of fruit at the time of harvest and add a colored photograph showing morphological differences among avocado fruits from studied cultivars. Since, all models have some limitations, it would be appropriate to add limitations for these models, too. Some English language improvements are suggested in the reviewed version of the manuscript.

Response #: We thank the review for constructive comments and suggestions to our manuscript. We added a colored photograph showing morphological differences of each avocado cultivars (Please see Fig 1 lines 106-107, and S1_Colored photograph of five Avocado cultivars.pdf, line 394 in the revised manuscript. In addition, we have added a recommendation to conduct a similar study using a large dataset to improve model performances and that can be used across regions (Please lines 249- 252, in the revised manuscript). We believe we were able to include all suggestions and comments including some language aspects in an adequate way.

---

## [Decision Letter · Decision Letter 1]

10 Jan 2022

PONE-D-21-36255R1Volume estimation models for avocado fruitPLOS ONE

Dear Dr. Mokria,

Thank you for submitting your manuscript to PLOS ONE. After careful consideration, we feel that it has merit but does not fully meet PLOS ONE’s publication criteria as it currently stands. Therefore, we invite you to submit a revised version of the manuscript that addresses the points raised during the review process.

We look forward to receiving your revised manuscript.

Kind regards,

Sajid Ali

Academic Editor

PLOS ONE

Journal Requirements:

Reviewers' comments:

**Reviewer #1: **

Thank you for your efforts in revising the manuscript in an appropriate manner. I believe that the revised manuscript has significant improvements. In other words, you answered all of the questions and made all of the necessary changes. In addition, the revised manuscript now includes significant and acceptable English language improvements, as well as a more in-depth discussion with adequate presentation and interpretation for each of the research findings.

Note that, regarding your responses #3 and #4 to reviewer #1, you must state your arguments in the main text of the manuscript as declarative sentences in order to inform readers. As a result, readers may be able to better understand the details of the reports and findings.

**Reviewer #2: **

Manuscript has been thoroughly reviewed. After review, the manuscript is in much better shape. Authors have properly incorporated or justified all suggested improvements.

**Academic Editor Comments**

There are two Hass cultivars in the figure 2 whereas in tables  the names are not same. Please be uniform in the whole manuscript about the names of the avocado cultivars. In addition, it would be better if the authors provide race information for the each cultivar. For your convenience, please refer to this publication i.e. https://doi.org/10.1016/j.scienta.2019.109008

---

## [Author Response · Author response to Decision Letter 1]

17 Jan 2022

Response to reviewers comment

ID: D-21-36255R1 

Title: Volume estimation models for avocado fruit

Dear Editor 

First, we would like to express our sincere appreciation to you and the anonymous reviewers for their time in reading the revised manuscript and providing constructive comments and suggestions which is relevant in improving the manuscript. Kindly find the response to each of the comments below. We believe we were able to address all suggestions and comments adequately, otherwise please contact us.

On behalf of all co-authors,

Sincerely,

Mulugeta Mokria

Response to Editor Comments

Comment #1: Please review your reference list to ensure that it is complete and correct. If you have cited papers that have been retracted, please include the rationale for doing so in the manuscript text, or remove these references and replace them with relevant current references. Any changes to the reference list should be mentioned in the rebuttal letter that accompanies your revised manuscript. If you need to cite a retracted article, indicate the article’s retracted status in the References list and also include a citation and full reference for the retraction notice.

Response #1: We are very grateful for your comment and suggestion. We checked the lists of the reference and they are complete and correct, please see lines 279-405 in the revised manuscript. In addition, in response to the reviewer's suggestion, we have cited a few more articles and changes are indicated in the revised manuscript. Please see lines, 39, 97, 279-283, 330, 403, in the revised manuscript. 

Response to reviewer #1

General comment. Thank you for your efforts in revising the manuscript in an appropriate manner. I believe that the revised manuscript has significant improvements. In other words, you answered all of the questions and made all of the necessary changes. In addition, the revised manuscript now includes significant and acceptable English language improvements, as well as a more in-depth discussion with adequate presentation and interpretation for each of the research findings.

General response: We are also grateful for your time in reading the revised manuscript and providing constructive comments and suggestions that considerably helped us to improve the manuscript. 

Comment #1: Note that, regarding your responses #3 and #4 to reviewer #1, you must state your arguments in the main text of the manuscript as declarative sentences in order to inform readers. As a result, readers may be able to better understand the details of the reports and findings.

Response #1: The reviewer makes a very important point, and we are grateful for that. As indicated by the reviewer, we added a statement that will guide readers to better understand the details of the type of model tested, and their performance statices, as well as the report in general. Please see lines 186-191, in the revised manuscript. 

Response to reviewer #2

General comment: Manuscript has been thoroughly reviewed. After review, the manuscript is in much better shape. Authors have properly incorporated or justified all suggested improvements.

General response: Thanks a lot and we are also grateful for your time in reading the revised manuscript and positive feedbacks. 

Response to Academic Editor Comments

Comment #1: There are two Hass cultivars in the figure 2 whereas in tables the names are not same. Please be uniform in the whole manuscript about the names of the avocado cultivars. In addition, it would be better if the authors provide race information for the each cultivar. For your convenience, please refer to this publication i.e. https://doi.org/10.1016/j.scienta.2019.109008

Response #1: Thanks a lot for your critical comments and suggestions to read an important article. In Fig 2, both are the same Hass cultivar. But, now we removed the replicated photos of the same cultivar to avoid further confusion and consistently used the same cultivar's name. Please see the revised Fig2 and the revised manuscript. Moreover, we further referred to the suggested article and provided “race” information for each cultivar. Please see lines 95-97 (in the main body) and lines 330-332 (in the reference list).

---

## [Editor Report · Decision Letter 2]

24 Jan 2022

Volume estimation models for avocado fruit

PONE-D-21-36255R2

Dear Dr. Mokria,

We’re pleased to inform you that your manuscript has been judged scientifically suitable for publication and will be formally accepted for publication once it meets all outstanding technical requirements.

Kind regards,

Sajid Ali

Academic Editor

PLOS ONE
---

## [Editor Report · Acceptance letter]

26 Jan 2022

PONE-D-21-36255R2 

Volume estimation models for avocado fruit 

Dear Dr. Mokria:

I'm pleased to inform you that your manuscript has been deemed suitable for publication in PLOS ONE. Congratulations! Your manuscript is now with our production department. 

Kind regards, 

on behalf of

Dr. Sajid Ali 

Academic Editor

PLOS ONE